# Neutrophil-to-Lymphocyte Ratio and Cytokine Profiling as Predictors of Disease Severity and Survival in Unvaccinated COVID-19 Patients

**DOI:** 10.3390/vaccines12080861

**Published:** 2024-07-31

**Authors:** Miguel Leonardo Méndez Rodríguez, Alberto Ponciano-Gómez, Myriam Campos-Aguilar, Wilfrido David Tapia-Sánchez, Carlos Leonardo Duarte-Martínez, Jesús Salvador Romero-Herrera, Sandra Olivas-Quintero, Alberto Daniel Saucedo-Campos, Adolfo Rene Méndez-Cruz, Rafael Jimenez-Flores, Vianney Ortiz-Navarrete, Hector Romero-Ramírez, Leopoldo Santos-Argumedo, Victor Hugo Rosales-García

**Affiliations:** 1Servicio de Inmunología y Alergia, Centro Médico Naval (CEMENAV), Secretaria de Marina (SEMAR), Avenida Heroica Escuela Naval Militar 745, Coapa, Presidentes Ejidales 1ra Sección, Coyoacán, Mexico City 04470, Mexico; esmenav.edu@gmail.com (M.L.M.R.); escposnav.jromero@uninav.edu.mx (J.S.R.-H.); 2Laboratorio de Inmunología (UMF), Facultad de Estudios Superiores Iztacala, Universidad Nacional Autónoma de México, Los Barrios N° 1, Los Reyes Iztacala, Tlalnepantla 54090, Estado de México, Mexico; alberto_ponciano@comunidad.unam.mx (A.P.-G.); myriam.campos@iztacala.unam.mx (M.C.-A.); alberto.saucedo@iztacala.unam.mx (A.D.S.-C.); armendez@unam.mx (A.R.M.-C.); jrjf@unam.mx (R.J.-F.); 3Diagnóstico Molecular de Leucemias y Terapia Celular (DILETEC), Basiliso Romo Anguiano 124, Industrial, Gustavo A. Madero, Mexico City 07800, Mexico; wtapia@diletec.com.mx (W.D.T.-S.); cduarte@diletec.com.mx (C.L.D.-M.); 4Departamento de Ciencias de la Salud Culiacán, Universidad Autónoma de Occidente, Culiacan 80020, Sinaloa, Mexico; sandra.olivas@uadeo.mx; 5Departamento de Biomedicina Molecular, Centro de Investigación y de Estudios Avanzados del Instituto Politécnico Nacional (CINVESTAV-IPN), Mexico City 07360, Mexico; vortiz@cinvestav.mx (V.O.-N.); hromero@cinvestav.mx (H.R.-R.); lesantos@cinvestav.mx (L.S.-A.); 6Centro de Investigación Sobre el Envejecimiento, Centro de Investigación y de Estudios Avanzados del Instituto Politécnico Nacional (CINVESTAV-IPN), Mexico City 07360, Mexico; 7Laboratorios Nacionales de Servicios Experimentales, Centro de Investigación y de Estudios Avanzados del Instituto Politécnico Nacional, Mexico City 14330, Mexico

**Keywords:** COVID-19, SARS-CoV-2, neutrophil-to-lymphocyte ratio (NLR), cytokine profiling, cytokines, immunological biomarkers

## Abstract

Background: During the COVID-19 pandemic, identifying reliable biomarkers for predicting disease severity and patient outcomes in unvaccinated individuals is essential. This study evaluates the efficacy of key hematological markers, including leukocyte and neutrophil counts, Neutrophil-to-Lymphocyte Ratio (NLR), and cytokine profiles (IL-6, INF-γ, TNF-α, IL-17A, CCL2, and CXCL10) for predicting the necessity for mechanical ventilation and assessing survival probabilities. Methods: We conducted an in-depth analysis on a cohort of COVID-19 patients, emphasizing the relationship between NLR, cytokine profiles, and clinical outcomes, utilizing routine leukocyte counting and cytokine quantification by flow cytometry. Results: Elevated leukocyte and neutrophil counts, increased NLR, and significant cytokine elevations such as IL-6 and IL-10 were strongly associated with the need for mechanical ventilation, reflecting a pronounced systemic inflammatory response indicative of severe disease outcomes. Conclusion: Integrating hematological markers, particularly NLR and cytokine profiles, is crucial in predicting mechanical ventilation needs and survival in non-vaccinated COVID-19 patients. Our findings provide critical insights into the pathophysiology of COVID-19, supporting the development of more targeted clinical interventions and potentially informing future strategies for managing infectious disease outbreaks.

## 1. Introduction

The emergence of the COVID-19 pandemic, triggered by the SARS-CoV-2 coronavirus in late 2019, represented a watershed moment in global public health [1,2,3]. In the initial phases, before vaccine availability, health systems worldwide grappled with unprecedented challenges in managing a disease that exhibited a wide and often lethal clinical spectrum—this period, marked by a globally unvaccinated population, provided a unique chance to examine COVID-19 in its rawest form, unaffected by vaccine-derived immunity [4,5].

In the initial phase of the pandemic, before widespread vaccination, the host’s immune response, especially the phenomenon known as ‘cytokine storm’, was recognized as a crucial determinant in progressing to severe forms of COVID-19 [6,7]. This response, marked by an excessive production of proinflammatory cytokines, led to severe complications, including respiratory and multiorgan failure [8,9,10].

Before the advent of vaccination campaigns, medical facilities across the globe were overwhelmed with cases, many exhibiting unpredictable progressions. This situation underscored the urgent need to identify clinical markers for early prediction of disease severity, aiming to optimize healthcare resources and enhance patient outcomes [11,12,13]. Consequently, the scientific community embarked on an intensive quest to decipher the pathogenesis of COVID-19, particularly focusing on the varied responses and disease severity among patients [14,15,16,17].

Given the pandemic’s overwhelming impact before the widespread implementation of vaccination campaigns, the need to identify reliable biomarkers for early-stage disease prediction became critical. The leukocyte count, specifically the neutrophil-to-lymphocyte ratio (NLR), emerged as a potential indicator of inflammation and disease severity [18,19,20]. However, the real-world clinical utility of these markers, especially in an unvaccinated cohort, warranted further exploration [21,22,23].

Moreover, the analysis of key cytokines such as IL-6, INF-γ, TNF-α, and IL-17A, alongside chemokines like CCL2 and CXCL10, provided invaluable insights into patients’ immunological status and potential clinical pathways. In a pre-vaccination era, understanding the role of these biomarkers was critical not only for grasping the virus’s virulent nature but also for shaping effective therapeutic strategies [24,25].

To be conducted amidst an active pandemic and focusing on an unvaccinated demographic, this study explores the predictive value of the Neutrophil-to-Lymphocyte Ratio (NLR) and cytokine profiling in predicting the severity and clinical outcomes of COVID-19 patients. Our main objective was to determine whether the NLR and specific cytokines could serve as reliable biomarkers for the early prediction of disease severity. The findings demonstrate that elevated NLR and specific cytokine profiles are significantly associated with adverse clinical outcomes, providing a valuable tool for patient risk stratification. Despite the evolving landscape of the pandemic, notably with the widespread adoption of vaccines, the insights gained from this research remain critically important. They offer an essential understanding of the immune response and the virus’s initial pathogenesis, knowledge that is imperative for managing future emergent infectious diseases and potential resurgences of SARS-CoV-2 variants.

## 2. Materials and Methods

### 2.1. Study Population and Sampling

This longitudinal observational study was conducted during the year 2020 at the Military Naval Hospital of Mexico, targeting unvaccinated adults hospitalized after initial evaluation for SARS-CoV-2 infection, confirmed via PCR. All individuals who met the inclusion criteria and were willing to participate signed the informed consent, forming a cohort of 323 patients with a demographic distribution of 123 females and 198 males, chosen to reflect the diversity of the hospitalized COVID-19 patient population. Approval was granted by the Ethical Committee of the Universidad Nacional Autónoma de México, Facultad de Estudios Superiores Iztacala (CE/FESI/022021/1380). Of these, 161 were intubated based on criteria including respiratory fatigue and other severe clinical markers, while 162 were not intubated. Additionally, 220 patients survived, and 103 patients did not survive. The average duration of hospital stay was 21.3 days, reflecting the prolonged care required for severe cases. Exclusion criteria included individuals with previous COVID-19 infection or those vaccinated against SARS-CoV-2. Intubation was considered for patients exhibiting signs of respiratory fatigue, a respiratory rate of 30–35 breaths per minute, SpO2 less than 90% despite oxygen therapy, PaO2 less than 60 mmHg, progressive hypercapnia with PaCO2 greater than 50 mmHg and pH less than 7.25, altered state of consciousness, and progressive hypotension and hypoperfusion. Details on the comorbidities of intubated and non-intubated patients, as well as surviving and deceased patients, are presented in Appendix A. Throughout their hospitalization, patients were subjected to weekly follow-up, including routine laboratory analyses and plasma cytokine quantification, until they were either successfully released from the hospital or deceased.

### 2.2. Cytokine Quantification by Flow Cytometry

A panel of cytokines including IL-1β, IL-2, IL-4, IL-6, IL-10, IL-12p70, IL-17A, CXCL8 (IL-8), CXCL10 (IP-10), CCL2 (MCP-1), IFN-γ, TNF-α, and TGF-β1 was measured in peripheral blood plasma. The LEGENDplex™ HU Essential Immune Response Panel (13-plex) kits from Biolegend (Biolegend, San Diego, CA, USA) were used following the manufacturer’s instructions to determine absolute cytokine levels. Data acquisition was performed on a CytoFLEX S (Beckman Coulter, Brea, CA, USA) flow cytometer, with LEGENDplex™ software (https://www.biolegend.com/) used for data analysis.

### 2.3. Peripheral Blood Cell Quantification

Routine counting of peripheral blood leukocyte populations, including lymphocytes, neutrophils, basophils, eosinophils, and monocytes, was performed using a Cell-Dyn Emerald hemocytometer (Abbott, Santa Clara, CA, USA). The neutrophil-to-lymphocyte ratio (NLR) was calculated from these data to indicate systemic inflammation.

### 2.4. Statistical Analysis

To evaluate the utility of the NLR, Receiver Operating Characteristic (ROC) curves were generated, analyzing values at admission, the highest peak, and the last measurement before discharge or death. R statistical software (https://www.r-project.org/) and pROC packages were used for these analyses. Comparisons of cytokine levels and peripheral blood cell counts, including NLR, across different patient groups were performed using Welch *t*-test. Statistical significance was determined based on a 95% confidence interval and *p* values ≤ 0.05.

## 3. Results

### 3.1. Dynamic Changes in Cytokine Levels and Neutrophil-to-Lymphocyte Ratio throughout COVID-19 Hospitalization

During sampling, points showing differences in the evaluated parameters correspond to the time of hospital admission, called ‘admission’. These points represent the values at the first sample collection within the health institution. The next important phase was when the highest value was reached for each parameter during the follow-up period. This moment will be called ‘Higher’ and followed by the parameter’s name, like ‘Higher NLR’ for the highest value in the neutrophil–lymphocyte ratio. It is important to note that this point will differ for each parameter and patient, representing the highest level recorded during hospitalization. The last evaluation for each parameter was one of the points in the patient follow-up where significant differences were found. This corresponds to the sample taken just before the patient was either successfully released from the hospital or before death. This point will be referred to as the ‘endpoint’ throughout the text, accompanied by the parameter designation, such as ‘Neutrophil endpoint’ for the final neutrophil value recorded in that particular patient.

### 3.2. Elevated Leukocyte, Neutrophil Counts, and NLR as Indicators of Intubation in COVID-19 Patients

Statistically significant differences in leukocyte and neutrophil counts and NLR were observed between patients requiring intubation and those who did not. Patients needing intubation showed higher counts of leukocytes and neutrophils at final and peak levels during hospitalization (Figure 1, upper panel). NLR values were also higher in intubated patients at admission, during the endpoint, and at the highest point (Figure 1, lower panel).

### 3.3. Assessing NLR Cutoff Points for Predicting Intubation in COVID-19 Patients

Of all hematological parameters evaluated, only the NLR at its three stages (admission, endpoint, and highest point) showed significant predictive capabilities for intubation, unlike the other hematological parameters. Optimal cutoff points for NLR varied in their predictive capabilities for intubation. At admission, a cutoff of ≥12.0 demonstrated a sensitivity of 75.8% and a specificity of 31.7%. For the NLR at the endpoint and the highest NLR, a cutoff of ≥10.7 resulted in sensitivities of 85.1% and 81.4% and specificities of 20.8% and 22.3%, respectively, (Figure 2B). Furthermore, intubated patients showed higher cytokine levels at the highest NLR, with average IL-6 levels of 736.29 pg/mL, INF-g of 95.45 pg/mL, TNF-a of 1061.33 pg/mL, IL17a of 10.04 pg/mL, CCL2 of 337.87 pg/mL, and CXCL10 of 487.78 pg/mL, compared to non-intubated patients as shown in Figure 2C.

### 3.4. Differential Leukocyte and Neutrophil Counts and NLR Dynamics in COVID-19 Patient Survival

Significant differences were noted when comparing cellular populations and the Neutrophil-to-Lymphocyte Ratio at various points during hospitalization between COVID-19 survivors and non-survivors. Non-survivors exhibited higher average leukocyte counts than survivors, evident in the final and highest values recorded during hospitalization (Figure 3, upper panel). Similarly, non-survivors also showed higher neutrophil counts. Regarding NLR, non-survivors presented higher values at admission, subsequent evaluations, and peak levels (Figure 3, lower panel).

### 3.5. Predictive Significance of NLR and Cytokine Levels in COVID-19 Survival

In the analysis of predictive significance for survival, only the NLR showed significant predictive capabilities. ROC analysis revealed variations in the predictive capacity of NLR. At admission, NLR had an AUC of 70.6%, with an optimal cut-off point of 8.6, indicating moderate specificity for non-survivors (76.2%) and survivors (46.11%) (Figure 4A and Figure 2B). The NLR at discharge and the highest recorded NLR demonstrated superior predictive abilities (AUCs of 90.7% and 89.9%, respectively), suggesting better mortality discrimination at these stages (Figure 4B). Additionally, significant differences were observed in cytokine levels at the highest NLR, with non-survivors showing an average IL-6 concentration of 1123.6 pg/mL and IL-10 concentration of 39.2 pg/mL, compared to 95.18 pg/mL and 11.08 pg/mL in survivors, respectively (Figure 4C).

## 4. Discussion

Understanding the interconnections between hematological markers, cytokine profiles, and clinical outcomes is pivotal in the complex landscape of COVID-19. Our study provides compelling evidence that elevated leukocyte and neutrophil counts and an increased Neutrophil-to-Lymphocyte Ratio are significant indicators for intubation in COVID-19 patients. Additionally, data from Appendix A may suggest that the differences observed in intubation status and survival are not primarily driven by the comorbidities listed, as no significant disparities were found between intubated and non-intubated patients or between surviving and deceased patients.

To enhance the discussion on the relationship between inflammatory markers and the severity of COVID-19, it is insightful to consider the neutrophil-to-lymphocyte ratio not only as an acute phase indicator but also as a reflection of the systemic inflammatory state, which may correlate with the progression to severe disease outcomes. The diagnostic and prognostic significance of the NLR in COVID-19 has been elucidated in a systematic review and meta-analysis, which concluded that an elevated NLR is associated with adverse outcomes in COVID-19 patients [21]. Recent studies (NLR1, NLR2) have further confirmed that an elevated NLR at hospital admission can predict the need for mechanical ventilation and is associated with higher mortality rates in unvaccinated COVID-19 patients [26,27,28]. These findings contribute to the expanding body of literature suggesting that an elevated NLR may be a readily accessible and cost-effective indicator to aid clinical decisions, particularly when considering the timing for intubation [29].

Moreover, in the context of systemic inflammatory responses, our findings agree with studies conducted in other inflammatory and autoimmune conditions. NLR has been utilized to indicate disease activity and treatment response [30]. For instance, in patients undergoing autologous hematopoietic stem cell transplantation for multiple sclerosis, a high NLR post-transplantation was linked to poorer outcomes [30]. These parallel results suggest that the mechanisms driving inflammation in COVID-19 could share common pathways with other inflammatory diseases, thus broadening the applicability of NLR as a universal indicator of disease severity.

The marked elevation in leukocyte and neutrophil counts at both the final and high levels during hospitalization in patients requiring intubation signifies a heightened state of inflammation. This aligns with the established role of IL-6 as a key predictor of the need for mechanical ventilation, as indicated by its strong association with respiratory failure and intubation [31,32]. The association of IL-6 with the need for mechanical ventilation also aligns with its prognostic implications in other infectious and non-infectious conditions. Elevated IL-6 levels have been observed in patients with severe disease states, not only in COVID-19 but also in conditions characterized by systemic inflammation, where IL-6 inhibition is beneficial [21,30]. This highlights the potential therapeutic implications of targeting IL-6 in managing COVID-19, where mitigating the cytokine storm could improve patient outcomes.

Furthermore, our observation of IL-6’s predictive capacity for intubation necessity finds support from broader immunological research, where IL-6 has been identified as a driver of the transition from innate to adaptive immunity, thus playing a crucial role in the pathogenesis of severe respiratory diseases, including COVID-19 [21,29]. These insights contribute to a more profound understanding of the inflammatory cascade in COVID-19 and reinforce the potential of NLR and IL-6 as biomarkers for guiding therapeutic strategies, particularly in the high-stakes decision-making process around intubation and ventilatory support.

Analyzing the cytokine profiles, we observed significant elevations in IL-6, INF-g, TNF-a, IL17a, CCL2, and CXCL10 at the highest NLR points in intubated patients. This cytokine surge not only underscores the disease’s severity but also suggests an overactive immune response, potentially contributing to the pathophysiology of COVID-19. The role of INF-g, identified as an independent risk factor associated with COVID-19 mortality [33], and the implications of IL17a in exacerbating immune responses and lung tissue damage are particularly complex but critical in understanding COVID-19’s immunopathology [34]. Elevated levels of IL-6, IL-10, and TNF-α have been shown to predict clinical progression to severe forms of COVID-19 [35,36]. This further corroborates the utility of these cytokines in assessing disease severity and guiding treatment decisions.

The role of TNF-a and IL17a in COVID-19 severity is particularly intriguing. TNF-a’s contribution to the severity of COVID-19 is well-documented, and its elevation in severe cases suggests a significant role in mediating the disease’s immunopathological processes [34,37,38]. IL17a, known for its role in promoting inflammation and tissue damage, emerges as a potential key player in the pathogenesis of COVID-19, particularly in severe cases [34,39]. Additionally, the elevation of these cytokines aligns with recent findings that link their levels to adverse outcomes in hospitalized patients [40].

Elevated levels of IL-6, INF-g, TNF-a, and IL17a mark an exacerbated inflammatory response and reflect the body’s attempt to combat the viral infection, with IL-6 being a pivotal mediator of fever and acute phase reactions [21]. The significant elevation of these cytokines in patients with higher NLR further corroborates the NLR’s utility as a marker of systemic inflammation and its potential to predict severe COVID-19 outcomes, aligning with findings that suggest a direct correlation between NLR and cytokine levels in determining disease severity [21,29].

The observation that cytokine levels are notably higher at the most elevated NLR points in intubated patients underlines the critical role of an overactive immune response in the progression to severe COVID-19, as detailed in the literature that posits NLR as a significant prognostic marker in the disease [21]. This cytokine surge, particularly the roles of INF-g and IL17a, highlights the dual nature of the immune response in COVID-19, where protective mechanisms against the virus may inadvertently contribute to tissue damage and disease exacerbation. The implications of such findings extend beyond diagnostic and prognostic considerations, hinting at potential therapeutic targets within the cytokine network to modulate the immune response favorably [29].

The significant roles of INF-g and IL17a, beyond their contributions to COVID-19 mortality and tissue damage, also underline the necessity of understanding the broader immunological landscape. For example, the systemic immune-inflammatory index (SII), which incorporates neutrophils, lymphocytes, and platelets, could offer a more holistic view of the inflammatory response in COVID-19 patients. Such an approach could help clarify the contributions of various cytokines to disease severity, offering insights into potential therapeutic targets [30].

Moreover, the elevation of TNF-a and IL17a in severe COVID-19 cases invites further investigation into their potential as biomarkers for predicting disease progression and as targets for therapeutic intervention. Their well-documented roles in inflammation and tissue damage make them prime candidates for targeted therapies to modulate the immune response to prevent severe outcomes [21,30].

On the other hand, CCL2 and CXCL10, elevated in severe cases, highlight the importance of chemokines in mediating immune responses and exacerbating lung injury in COVID-19 [41,42,43]. The elevation of these cytokines aligns with their roles in immune cell recruitment and activation, suggesting a direct involvement in the pathological processes underlying severe COVID-19.

Expanding upon the significance of chemokines such as CCL2 and CXCL10 in severe COVID-19 cases, it is critical to understand their function in orchestrating the immune response, especially in lung injury and systemic inflammation. The elevation of these chemokines not only signifies an aggravated immune response but also points to their crucial roles in recruiting and activating immune cells at sites of infection, which can exacerbate lung damage and contribute to the severity of COVID-19.

Moreover, the roles of CCL2 and CXCL10 extend beyond simple chemotaxis; they are intricately involved in the pathophysiological mechanisms that drive the progression of COVID-19 to its more severe forms. By mediating the infiltration of immune cells into lung tissue, these chemokines can initiate and perpetuate a cycle of inflammation and tissue damage, underscoring the importance of targeted therapeutic strategies aimed at modulating the chemokine response to mitigate severe disease manifestations [29].

Considering these insights, it is evident that targeting the chemokine axis could represent a promising therapeutic avenue. For instance, interventions designed to inhibit the signaling pathways of CCL2 and CXCL10 may help to reduce excessive immune cell recruitment and activation, thereby limiting inflammation and preventing the escalation of lung injury in COVID-19 patients. This approach aligns with the broader concept of using biomarkers, such as NLR and specific cytokine profiles, to guide therapeutic interventions and improve patient outcomes, as highlighted in recent research on using NLR as markers of disease severity and treatment response in autoimmune diseases [30].

The direct involvement of chemokines in the pathological processes of severe COVID-19 thus adds a layer to our understanding of the disease’s immunopathology. Future studies should further elucidate the roles of these and other chemokines in COVID-19, exploring their potential as biomarkers for disease severity and as targets for therapeutic intervention. Such research could pave the way for developing more effective treatments that address the underlying mechanisms of disease progression, ultimately improving the prognosis for patients with COVID-19 [21,29,30].

Considering these findings, the coexistence of elevated NLR and cytokine levels, particularly IL-6 and IL-10, offers a window into the complex dynamics of the immune response to COVID-19. The simultaneous increase in these markers suggests increased disease severity and provides insights into the dysregulated immune state characteristic of severe COVID-19 [44,45,46]. Recent studies have confirmed the critical role of IL-6 and IL-10 in predicting severe outcomes in hospitalized COVID-19 patients, further validating our findings [35,36].

The elevated levels of IL-10 in severe COVID-19 cases offer intriguing insights into the cytokine dynamics at play within the disease’s pathogenesis. Traditionally recognized for its anti-inflammatory properties, IL-10’s increased presence in severe instances of COVID-19 might reflect the body’s compensatory response to mitigate the disease’s rampant inflammation [21]. This adaptation underscores the complex interplay between pro-inflammatory and anti-inflammatory responses, with IL-10 as a critical mediator aiming to restore homeostasis and prevent immune-mediated damage. This is consistent with recent findings that highlight IL-10’s role in balancing the immune response in severe COVID-19 cases [40].

Furthermore, the nuanced role of IL-10 in COVID-19’s immunopathology can be paralleled with findings in other inflammatory conditions, where an elevated NLR, alongside heightened levels of pro-inflammatory cytokines, has been associated with adverse outcomes. This suggests that IL-10 elevation might not solely act as a protective mechanism but could indicate an overwhelmed immune system struggling to control an unbridled inflammatory response [21,30].

The significance of IL-10 in severe COVID-19 cases also prompts a reevaluation of therapeutic strategies aimed at modulating the immune response. Given its potential role in counterbalancing excessive inflammation, therapies that enhance IL-10 activity or mimic its effects could offer novel avenues for treatment, especially in preventing the progression to severe disease stages. This aligns with the growing interest in identifying biomarkers that can guide the development of targeted therapies, underscoring the need for further research to unravel the intricate balance of cytokine responses in COVID-19 and their implications for patient management [21,29,30].

In summary, our study underscores the predictive value of the Neutrophil-to-Lymphocyte Ratio (NLR) and cytokine profiles, particularly IL-6 and IL-10, as markers in COVID-19. This reflects the complexity of the immune response and its impact on disease severity. Our findings highlight the importance of a multimodal approach in the clinical assessment of patients, where integrating inflammatory markers can facilitate early identification of individuals at risk of progressing to severe complications.

The elevation of pro-inflammatory cytokines and IL-10, an anti-inflammatory cytokine, unveils the dual dynamics of the immune response in the pathogenesis of COVID-19, suggesting the need for therapeutic strategies that effectively modulate this response. Furthermore, comparing our results with findings in other inflammatory and autoimmune conditions emphasizes the potential universality of these markers as indicators of disease severity, extending their applicability beyond SARS-CoV-2 infection [21,29,30].

It is imperative to continue exploring the role of inflammatory markers and specific cytokines in COVID-19 and other diseases to develop targeted interventions that can mitigate disease progression and improve patient outcomes. This study contributes to the understanding of COVID-19 and lays the groundwork for future research to optimize the clinical management of this complex disease.

## 5. Conclusions

The neutrophil-to-lymphocyte ratio and cytokine profiling play a significant role in predicting disease severity and guiding clinical decisions in unvaccinated COVID-19 patients. Elevated NLR and cytokine levels, particularly IL-6 and IL-10, were strongly associated with the need for mechanical ventilation and poor outcomes, emphasizing their utility as biomarkers for early intervention and risk stratification.

These findings suggest that integrating NLR and cytokine profiles into routine clinical assessments can enhance patient management by identifying those at higher risk for severe disease. This approach may help optimize the timing of intubation and other critical interventions, ultimately improving patient outcomes.

Future research should aim to validate these biomarkers in diverse populations and explore their relevance in other infectious and non-infectious diseases. Additionally, exploring therapeutic strategies that modulate cytokine responses, especially to mitigate the cytokine storm in severe COVID-19 cases, remains crucial. The insights gained from this study contribute to a more profound understanding of COVID-19’s immunopathology and could be applicable in managing severe inflammatory responses in various clinical settings, beyond COVID-19.

## Figures and Tables

**Figure 1 vaccines-12-00861-f001:**
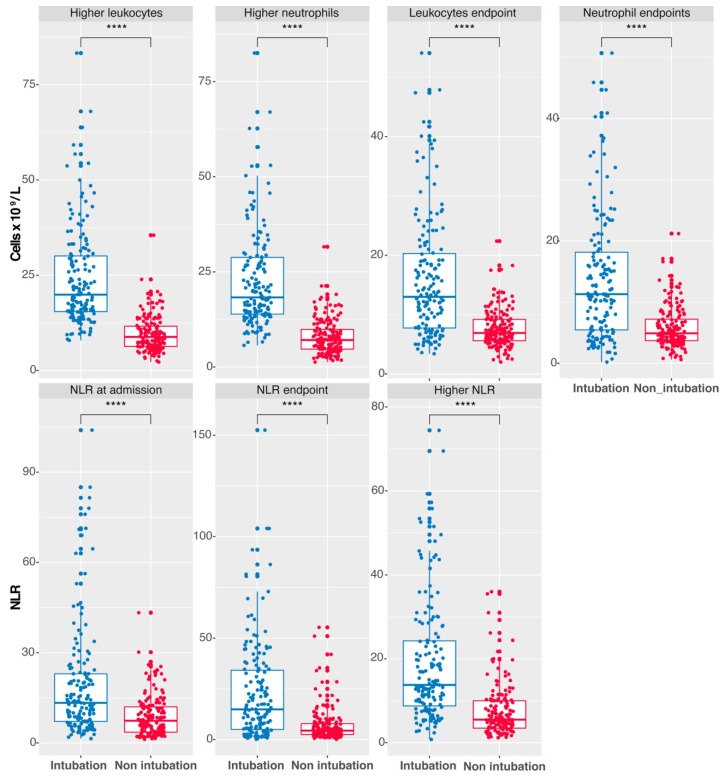
Hematologic Biomarkers Distinguishing Intubated from Non-intubated COVID-19 Patients. Distribution of leukocyte and neutrophil counts and NLR values compared intubated (in blue) to non-intubated (in red) COVID-19 patients. Admission: initial hospital values. Higher: maximum values reached. Endpoint: final values successfully released from the hospital or before the patient’s death. (**** *p* ≤ 0.0001, Welch *t*-test). Vertical lines show the standard error.

**Figure 2 vaccines-12-00861-f002:**
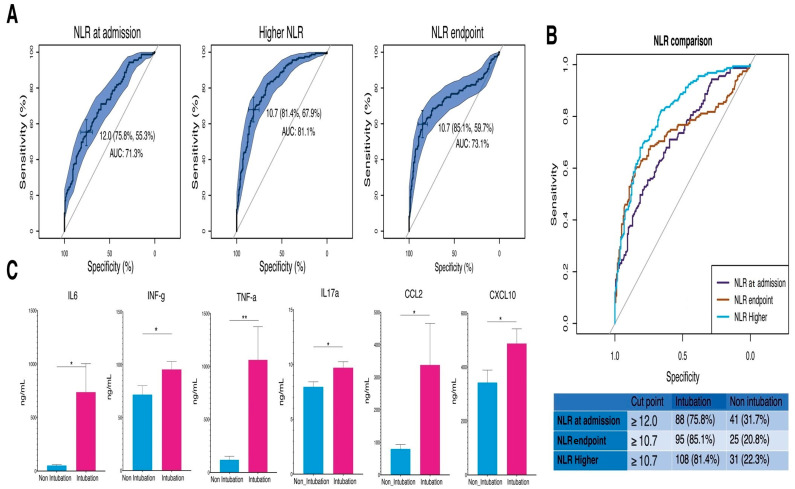
Analysis of NLR Cutoff Points and Cytokine Levels in Intubation Prediction. (**A**) ROC curves for NLR at admission, higher NLR during hospitalization, and NLR at endpoint, indicating their predictive accuracy for intubation with respective AUC percentages. (**B**) ROC curve comparison highlights the statistical distinction between the predictive values of NLR at different points. (**C**) Bar graphs showing elevated cytokine levels (IL-6, INF-γ, TNF-α, IL-17A, CCL2, CXCL10) at the highest NLR in intubated patients compared to non-intubated patients, underscoring the increased inflammatory response associated with the need for ventilatory support. Higher: maximum values reached. Endpoint: final values successfully released from the hospital or before the patient’s death. (* *p* ≤ 0.05, ** *p* ≤ 0.01, Welch *t*-test). vertical lines show the standard error.

**Figure 3 vaccines-12-00861-f003:**
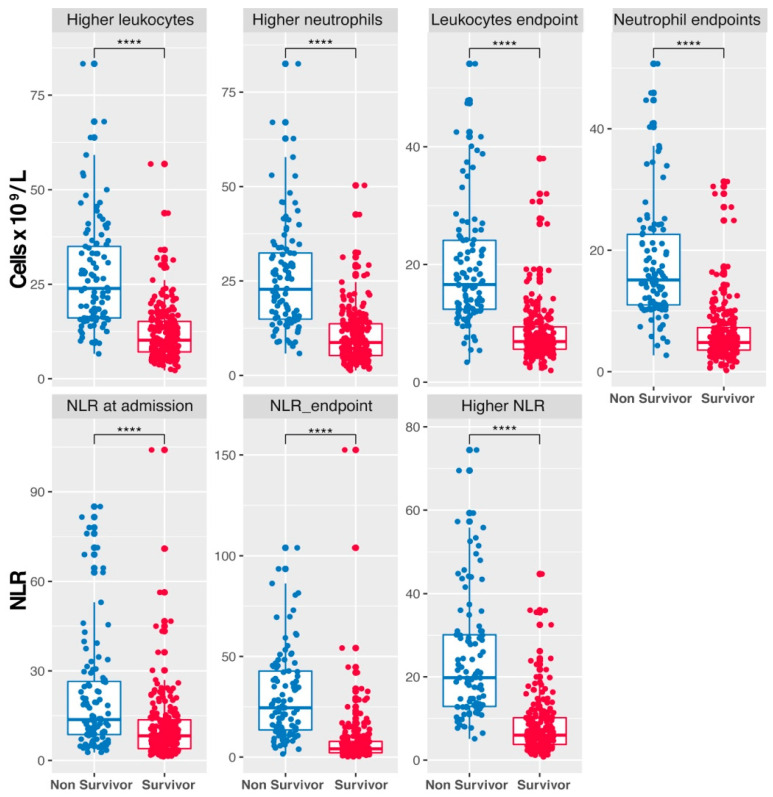
Hematologic biomarkers differentiating survivors from non-survivors in COVID-19 Patients. Box plots illustrating the distribution of leukocyte and neutrophil counts, along with Neutrophil-to-Lymphocyte Ratio (NLR) values, comparing non-survivors (in blue) to survivors (in red) of COVID-19. Higher counts and NLR values are associated with non-survival. Higher: maximum values reached. Endpoint: final values successfully released from the hospital or before the patient’s death. (**** *p* ≤ 0.0001, Welch *t*-test). Vertical lines represent the standard error.

**Figure 4 vaccines-12-00861-f004:**
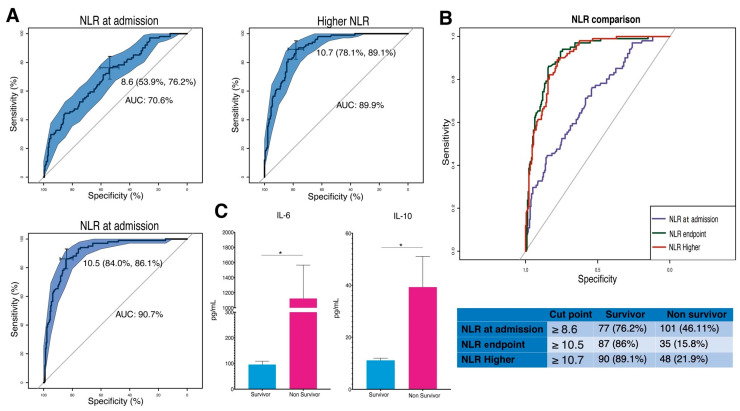
Assessing NLR and Cytokine Levels as Indicators of Survival in COVID-19. (**A**) ROC curves for NLR at admission, the highest recorded NLR, and NLR at endpoint, with AUC values indicating their discriminative power for patient survival. (**B**) Comparative analysis of ROC curves demonstrates NLR’s differential predictive effectiveness at various clinical stages. (**C**) Bar charts illustrate the levels of IL-10 and IL-6, compare survivors (in red) to non-survivors (in blue) at the highest NLR, and highlight the marked cytokine elevation associated with non-survival cases. Higher: maximum values reached. Endpoint: final values successfully released from the hospital or before the patient’s death. (* *p* ≤ 0.05, Welch *t*-test). Vertical lines show the standard error.

## Data Availability

Dataset available on request from the authors.

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
