# Peer review of "Neutrophil-to-Lymphocyte Ratio and Cytokine Profiling as Predictors of Disease Severity and Survival in Unvaccinated COVID-19 Patients"

_vaccines, 2024, doi:10.3390/vaccines12080861_

Round 1

Reviewer 1 Report

Comments and Suggestions for Authors

Sample

This is a very specific patient population chosen with these criteria:

Military Naval Hospital of Mexico

unvaccinated adults hospitalized after initial evaluation for SARS- 92 CoV-2 infection, confirmed via PCR

The study cohort consisted of 323 patients, with a demographic distribution of 123 females and 198 males, chosen to reflect the diversity of the hospitalized COVID-19 patient population.

Exclusion criteria included individuals with previous 97 COVID-19 infection or those vaccinated against SARS-CoV-2.

Intubation was considered of 13 for patients exhibiting signs of respiratory fatigue, a respiratory rate of 30-35 breaths per 99 minute, SpO2 less than 90% despite oxygen therapy, PaO2 less than 60mmHg, progressive hypercapnia with PaCO2 greater than 50 mmHg and pH less than 7.25, altered state of consciousness, and progressive hypotension and hypoperfusion.

[please explain how the 323 patients were selected. What was the size of the group of eligibles? What is the generalisability of the results for the 13 patients?]

Methods

Well described

Results

You wrote” ROC analysis revealed variations in the predictive capacity of NLR. At admission, NLR had an AUC of 70.6%, with an optimal cut-off point of 8.6, indicating moderate specificity for non-survivors (76.2%) and survivors (46.11%) (Figure 4A, 2B). The NLR at discharge and the highest recorded NLR demonstrated superior predictive abilities (AUCs of 90.7% and 89.9%, respectively), suggesting better mortality discrimination at these stages (Figure 4B). Additionally, significant differences were observed in cytokine levels at the highest NLR, with non-survivors showing an average IL-6 concentration of 1123.6 ng/mL and IL-10 concentration of 39.2 ng/mL, compared to 95.18 ng/mL and 11.08 ng/mL in survivors, respectively (Figure 4C).”

[These high AUC values are very suggestive. However, you provide no comparative data on other risk factors for the patients – medical comorbidities, age, smoking history…. Please state which other factors you would wish to control for in future].  

Conclusions

You wrote: “Future research should further validate these biomarkers across diverse populations and explore their utility in other infectious and non-infectious diseases. Additionally, studies should investigate the therapeutic implications of modulating cytokine responses, particularly focusing on interventions that can alleviate the cytokine storm associated with severe COVID-19 cases.”

[can you please provide data from the literature on how these biomarkers differ in “other infectious diseases”]

[Please state clearly whether you think the data from your 13 key patients can provide clinical guidance or how much more data are needed for clear clinical guidance]

Author Response

Comment 1:

Please explain how the 323 patients were selected. What was the size of the group of eligibles? What is the generalisability of the results for the 13 patients?

Response:

Thank you for your comments. We have added specific details about the number of intubated and non-intubated patients, as well as the number of surviving and deceased patients to the manuscript; this information is now found between lines 98 and 101. Regarding participant selection, we included all patients who met the inclusion criteria and were willing to participate during 2020, detailed between lines 91 and 93-94. This process ensured that we captured a representative sample of the eligible population.

Regarding the generalizability of our results, we clarify that our conclusions are informed by the comprehensive analysis of both intubated and non-intubated groups, which contributes to our understanding of the biomarkers in a clinical context.

Comment 2:

These high AUC values are very suggestive. However, you provide no comparative data on other risk factors for the patients – medical comorbidities, age, smoking history…. Please state which other factors you would wish to control for in future

Response:

Thank you for your comments. We have added two supplementary tables presenting the comorbidities of intubated and non-intubated patients (Supplementary Table 1) and surviving and deceased patients (Supplementary Table 2). These tables are specifically mentioned in lines 107-109 of the methods and lines 227-230 of the discussion of the manuscript. Additionally, a chi-squared analysis was performed, supporting the observation that there are no significant differences in comorbidities between the groups. The inclusion of these supplementary tables supports the observation that comorbidities do not predominantly influence intubation or survival status.

Comment 3:

can you please provide data from the literature on how these biomarkers differ in “other infectious diseases”

Response:

Thank you for your comments. We have conducted a comprehensive review of the literature to provide data on how the biomarkers mentioned in our study differ in other infectious diseases.

Elevated levels of the neutrophil-to-lymphocyte ratio (NLR) have proven to be a significant prognostic marker not only in COVID-19 but also in a variety of other infections. For example, NLR has been correlated with the severity of diseases such as sepsis and influenza, indicating its utility as a general indicator of inflammation and immune response in multiple infectious contexts (References 1-4).

Interleukin-6 (IL-6) has also been widely studied in various infections. Elevated levels of IL-6 have been associated with worse outcomes in patients with sepsis and pneumonia, which is consistent with our findings in COVID-19, underscoring its crucial role in the acute inflammatory response (References 5, 6).

Regarding interleukin-17 (IL-17), this cytokine has shown to be a vital component in the immune response against bacterial, fungal, and viral infections. Its involvement in the pathogenesis of diseases such as tuberculosis and candidiasis reflects its broad relevance in infections (References 9, 10).

Interferon-gamma (IFN-γ) is another key cytokine whose elevation has been observed in viral infections such as HIV and hepatitis C, indicating its central role in T-cell-mediated immunity and its importance in antiviral immune response (References 7, 8).

Monocyte chemoattractant protein 1 (CCL2) and interferon gamma-induced protein 10 (CXCL10) are chemokines that also play critical roles in attracting and activating immune cells during infections. Their elevation has been documented in infections such as dengue and malaria, supporting their use as biomarkers of immune activation in various infectious diseases (References 11, 12, 13, 14).

Finally, interleukin-10 (IL-10), an anti-inflammatory cytokine, has shown to have a modulatory role in infections such as sepsis and hepatitis B, highlighting its importance in regulating excessive inflammatory responses and its potential in therapeutic interventions (References 15, 16).

These findings reinforce the relevance of the biomarkers studied not only in COVID-19 but also across a wide range of infectious diseases, emphasizing the need to continue investigating their roles and therapeutic applications.

References:

  1. Russell, C. D., Parajuli, A., Gale, H. J., Bulteel, N. S., Schuetz, P., de Jager, C. P., ... & Baillie, J. K. (2019). The utility of peripheral blood leucocyte ratios as biomarkers in infectious diseases: A systematic review and meta-analysis. Journal of Infection, 78(5), 339-348.
  2. Buonacera, A., Stancanelli, B., Colaci, M., & Malatino, L. (2022). Neutrophil to lymphocyte ratio: an emerging marker of the relationships between the immune system and diseases. International journal of molecular sciences, 23(7), 3636.
  3. Prozan, L., Shusterman, E., Ablin, J., Mitelpunkt, A., Weiss-Meilik, A., Adler, A., ... & Kehat, O. (2021). Prognostic value of neutrophil-to-lymphocyte ratio in COVID-19 compared with Influenza and respiratory syncytial virus infection. Scientific reports, 11(1), 21519.
  4. Drăgoescu, A. N., Pădureanu, V., Stănculescu, A. D., Chiuțu, L. C., Tomescu, P., Geormăneanu, C., ... & Drăgoescu, O. P. (2021). Neutrophil to lymphocyte ratio (NLR)—a useful tool for the prognosis of sepsis in the ICU. Biomedicines, 10(1), 75.
  5. Cong, S., Ma, T., Di, X., Tian, C., Zhao, M., & Wang, K. (2021). Diagnostic value of neutrophil CD64, procalcitonin, and interleukin-6 in sepsis: a meta-analysis. BMC infectious diseases, 21, 1-17.
  6. Liu, S., Wang, X., She, F., Zhang, W., Liu, H., & Zhao, X. (2021). Effects of neutrophil-to-lymphocyte ratio combined with interleukin-6 in predicting 28-day mortality in patients with sepsis. Frontiers in immunology, 12, 639735.
  7. Greco, M., Mazzei, A., Suppressa, S., Palumbo, C., Verri, T., & Lobreglio, G. (2021). Human leukocyte antigen-DR isotype expression in monocytes and T cells interferon-gamma release assay in septic patients and correlation with clinical outcome. Journal of clinical medicine research, 13(5), 293.
  8. Kak, G., Raza, M., & Tiwari, B. K. (2018). Interferon-gamma (IFN-γ): Exploring its implications in infectious diseases. Biomolecular concepts, 9(1), 64-79.
  9. Sahu, U., Biswas, D., Prajapati, V. K., Singh, A. K., Samant, M., & Khare, P. (2021). Interleukin‐17—A multifaceted cytokine in viral infections. Journal of Cellular Physiology, 236(12), 8000-8019.
  10. Retnoningrum, D., Sofro, M. A. U., Pramudo, S. G., Wicaksono, S. A., & Dahesihdewi, A. (2022). Correlation between Interleukin-17 Levels with C-Reactive Protein and Neutrophil Lymphocyte Ratio in Sepsis. Journal of Biomedicine and Translational Research, 8(2), 86-90.
  11. Den Hartog, G., Ederveen, T. H. A., Venkatasubramanian, P. B., Ferwerda, G., van den Kieboom, C. H., van der Gaast-de Jongh, C. E., ... & de Jonge, M. I. (2021). Chemokine profiling in children and adults with symptomatic and asymptomatic respiratory viral infections. Journal of Infection, 83(6), 709-737.
  12. Zhu, J., Liu, X., Zhan, X., Wang, M., Zhang, Y., Na, L., & Li, S. (2023). Predictive value of chemokines (CCL 2) in bronchoalveolar lavage fluid for refractory mycoplasma pneumonia in children. Italian Journal of Pediatrics, 49(1), 125.
  13. Li, M., Chen, Y., Li, H., Yang, D., Zhou, Y., Chen, Z., & Zhang, Y. (2021). Serum CXCL10/IP-10 may be a potential biomarker for severe Mycoplasma pneumoniae pneumonia in children. BMC Infectious Diseases, 21, 1-8.
  14. Elemam, N. M., Talaat, I. M., & Maghazachi, A. A. (2022). CXCL10 chemokine: a critical player in RNA and DNA viral infections. Viruses, 14(11), 2445.
  15. Herawati, S., Kandarini, Y., Mulyantari, N. K., & Prabawa, P. Y. (2022). Correlation of Neutrophil to Lymphocyte Ratio with Interleukin-10 in Diagnosis and Monitoring of Coronavirus Disease-19 Patients. Open Access Macedonian Journal of Medical Sciences, 10(B), 63-66.
  16. König, R., Kolte, A., Ahlers, O., Oswald, M., Krauss, V., Roell, D., ... & Bauer, M. (2021). Use of IFNγ/IL10 ratio for stratification of hydrocortisone therapy in patients with septic shock. Frontiers in immunology, 12, 607217.

Reviewer 2 Report

Comments and Suggestions for Authors

On May 5, 2023, the head of WHO declared an end to the global health emergency caused by COVID-19. However, the problem of COVID-19 is still relevant today. This concerns both the severe complications of acute COVID-19 and the long-term consequences of the disease. At the same time, the importance of vaccination in preventing these complications, especially those associated with systemic inflammation, is still being debated. The authors found a strong prognostic role of cytokines and chemokines, as well as the neutrophil/lymphocyte ratio, in relation to critical complications in previously unvaccinated patients. Similar studies have been performed previously, but they did not clearly distinguish patients who did not receive vaccines. Therefore, this work has some practical relevance and may be of interest to Journal Vaccines readers. In the meantime, I have a few comments:

(1) In the text, including the abstract, there are: inaccurate spelling of cytokines: INF-g, TNF-a, IL17a, should read: INF-γ, TNF-α and IL-17A.

(2) 2.1. Study population and sampling. It is desirable to indicate the period during which the study was conducted.

(3) 2.4. Statistical analysis. The use of Student's t-test is correct only if the distribution is normal. It is necessary to state how the normality of the distribution was checked and by what method.

(4) 3. Results. Cytokines are measured in pg/mL, not ng/mL.

(5) 5. Conclusions. It is desirable to be more concise and focus on the main results.

Author Response

Comment 1:

In the text, including the abstract, there are: inaccurate spelling of cytokines: INF-g, TNF-a, IL17a, should read: INF-γ, TNF-α and IL-17A.

Response:

Thank you for your comments. We have corrected the spelling of the cytokines in the abstract. The corrected nomenclature is now found on line 31 of the revised manuscript.

Comment 2:

Study population and sampling. It is desirable to indicate the period during which the study was conducted.

Response:

Thank you for your comments. We have added the period during which the study was conducted to the manuscript. This information is now included in line 91 of the revised manuscript.

Comment 3:

2.4. Statistical analysis. The use of Student's t-test is correct only if the distribution is normal. It is necessary to state how the normality of the distribution was checked and by what method.

Response:

Thank you for your comment. I apologize for the error in the manuscript. It was my mistake to write Student's t-test; in fact, the Welch t-test was used from the beginning as the normality tests, including the Shapiro-Wilk test, showed that the data distribution was not normal. This correction has been made in the manuscript, specifically in the figure legends and in line 132. attached are the p-values of the analysis.

Hematologic biomarkers

T Statistic

p-value

Intubated versus Non-intubated

Higher leukocytes

13.45

1.01E-29

Higher neutrophils

13.63

2.73E-30

Leukocytes endpoint

9.74

2.03E-18

Neutrophil endpoints

9.40

1.76E-17

NLR at admission

6.53

5.41E-10

NLR endpoint

7.67

7.85E-13

Higher NLR

9.13

4.85E-17

Survivor versus Non-survivor

Higher leukocytes

10.09

7.91E-18

Higher neutrophils

10.48

8.77E-19

Leukocytes endpoint

10.53

8.73E-19

Neutrophil endpoints

11.38

6.77E-21

NLR at admission

4.98

1.89E-06

NLR endpoint

9.69

2.53E-17

Higher NLR

10.47

1.41E-18

Cytokine

T Statistic

p-value

Intubation versus Non-Intubation

IL-6

2.57

0.0116

INF-γ

2.11

0.0361

TNF-⍺

2.98

0.0035

IL17A

2.64

0.009

CCL2

2.00

0.0479

CXCL10

2.03

0.0435

Survivor versus Non-Survivor

IL-6

-2.32

0.0202

IL-10

-2.38

0.0234

Comment 4:

Results. Cytokines are measured in pg/mL, not ng/mL.

Response:

Thank you for your comment. We appreciate your attention to detail. The error originated from the scale of the standard curve in the kit, which displays ng/mL, and the software provides values without units. After confirming with the supplier that the software outputs values in pg/mL, we have corrected the units in Figures 2 and 4 accordingly. Additionally, these changes have been reflected in the text on lines 173-175 and 211-212 to ensure accuracy in the reported data.

Comment 5:

Conclusions. It is desirable to be more concise and focus on the main results.

Response:

Thank you for your valuable feedback. We have revised the conclusion to be more concise and focused on the main results. The updated conclusion now emphasizes the significant role of NLR and cytokine profiling in predicting disease severity and guiding clinical decisions in unvaccinated COVID-19 patients, and it highlights the potential application of these biomarkers in other infectious and non-infectious diseases. This revision can be found in lines 387 to 402.

Reviewer 3 Report

Comments and Suggestions for Authors

Considering the apparent interest of the NLR ratio despite a low PPV (but a high NPV) and the apparent  clinical value of IL-6, TNF and IL-10 dosing their relationship with the patient status should be further investigated.

The values of the different markers are somewhat surprising as compared to other studies and the complete data should be presented (as in figure one).

A multivariate analysis might be of interest.

The authors declare to have performed weekly evaluations of the biological parameters and clinical data of patients and unfortunately they use only partially these data. What is the mean duration stay of the patients in the hospital? Is it possible to have a description of the evolution of these parameters by groups of patients from hospital admission until their discharge (or death)? That would give insight of the relation between the most relevant parameters (among the ones used by the authors) and the outcome of the patients (over several weeks?). It would be useful (although the study is limited to one center) for their follow-up.

The addresses of the companies which provided the reagents and the laboratory equipment have to be indicated.

Author Response

Comment 1:

Considering the apparent interest of the NLR ratio despite a low PPV (but a high NPV) and the apparent clinical value of IL-6, TNF, and IL-10 dosing, their relationship with the patient status should be further investigated.

Response:

We appreciate your insightful thoughts. In this initial dataset, we tested the correlation of cytokines alone with various clinical parameters beyond what has already been reported in this version. However, we did not find significant correlations, leading us to report only the significant findings. These results suggest that there may be deeper interactions between these cytokines and the pathophysiological processes, particularly those affecting lung tissue. To further explore these aspects and correlate these cytokines with specific pathological mechanisms, especially in patients who succumbed to the disease, for whom tissue samples are available for analysis, we are currently conducting a continuous follow-up study on COVID-19 patients.

We believe that this continuing research will provide a more comprehensive understanding of the complex relationships between cytokines and clinical outcomes in COVID-19.

Comment 2:

The values of the different markers are somewhat surprising as compared to other studies and the complete data should be presented (as in figure one).

Response:

Thank you for your valuable comments. In response to your suggestion, we have clarified in the manuscript the rationale for presenting only the significant results. Specifically, we have mentioned that of all hematological parameters evaluated, only the NLR at its three stages (admission, endpoint, and highest point) showed significant predictive capabilities for intubation, unlike the other hematological parameters. This information has been added to the text in lines 166-168. Similarly, we clarified that in the analysis of predictive significance for survival, only the NLR showed significant predictive capabilities, which has been added in lines 203-204.

Comment 3:

A multivariate analysis might be of interest.

Response:

Thank you for your insightful comment. We initially performed a Principal Component Analysis (PCA) as an approach to multivariate analysis. However, the results indicated that for the conditions of intubation and survival, the primary components for these groups were the NLR in the three stages analyzed (admission, endpoint, and highest point) as well as all the evaluated cytokines. Therefore, we proceeded with the analyses focused on these significant parameters.

Comment 4:

The authors declare to have performed weekly evaluations of the biological parameters and clinical data of patients and unfortunately they use only partially these data. What is the mean duration stay of the patients in the hospital? Is it possible to have a description of the evolution of these parameters by groups of patients from hospital admission until their discharge (or death)? That would give insight into the relation between the most relevant parameters (among the ones used by the authors) and the outcome of the patients (over several weeks?). It would be useful (although the study is limited to one center) for their follow-up.

Response:

Thank you for your comments. We have added the average duration of hospital stay to the manuscript, which is now found in lines 101 to 102. The mean duration of hospital stay was 21.3 days, reflecting the prolonged care required for severe cases. We agree that a longitudinal analysis of the evolution of parameters over the course of hospitalization would provide valuable insights. This approach is indeed interesting and is part of our ongoing research. Due to the complexity and volume of the data, these analyses are taking longer to complete, and we aim to present them in future publications as we gather more comprehensive data. This ongoing work focuses on correlating cytokine levels and other biomarkers with clinical outcomes over time, which we believe will contribute significantly to understanding the disease's progression and improving patient management. The initial analysis of longitudinal data suggests that a detailed examination of these trends will provide deeper insights, which we are currently pursuing.

Comment 5:

The addresses of the companies which provided the reagents and the laboratory equipment have to be indicated.

Response:

Thank you for your observation. We have added the addresses of the companies that provided the reagents and laboratory equipment. This information can be found in the methods section between lines 118, 120, and 125.

Round 2

Reviewer 1 Report

Comments and Suggestions for Authors

Thanks to the authors for their detailed replies to the reviewer's comments. They have made appropriate changes within the limits of their sample size, and in the discussion section which are very helpful.

Author Response

Comment:

Thanks to the authors for their detailed replies to the reviewer's comments. They have made appropriate changes within the limits of their sample size, and in the discussion section which are very helpful.

Response:

Thank you for your positive feedback. We appreciate your thorough review, which has helped us improve specific points in the manuscript. Your suggestions have been very valuable in enhancing the clarity and depth of our discussion.

Reviewer 3 Report

Comments and Suggestions for Authors

The paper is well written and data are clearly presented. Closely related results have, at the present time, already been described in several recent studies and should be taken into account.

References should be updated: very recent papers have been published in 2024 (in open access) have evaluated the prognosis markers used by the authors in the hospitalized patients for COVID-19.

Considering the apparent interest of the NLR ratio despite a low PPV (but a high NPV) and the apparent clinical value of IL-6, TNF and IL-10 dosing their relationship with the patient status should be further investigated.

The values of the different marker are somewhat surprising as compared to other studies and the complete data should be presented (as in figure one).

A multivariate analysis might be of interest.

The authors declare to have performed weekly evaluations of the biological parameters and clinical data of patients and unfortunately they use only partially these data. What is the mean duration stay of patients in the hospital? Is it possible to have a description of the evolution of these parameters by groups of patients from hospital admission until their discharge (or death)? That would give insight of the relation between the most relevant parameters (among the ones used by the authors) and the outcome of the patients (over several weeks?). It would be useful (although the study is limited to one center) for their follow-up.

The addresses of the companies which provided the reagents and the laboratory equipment have to be indicated.

Author Response

Comment 1:

References should be updated: very recent papers have been published in 2024 (in open access) have evaluated the prognosis markers used by the authors in the hospitalized patients for COVID-19.

Response:

We have conducted a thorough search for recent literature from 2024 that correlates the parameters we evaluated in COVID-19 patients. As a result, we have incorporated additional references in the discussion section, now found on lines 238 to 241, 278 to 280, 286 to 287, 352 to 354, and 361 to 363. Furthermore, we have added new references (26, 27, 28, 35, 36, and 40) to the bibliography.

Comment 2:

Considering the apparent interest of the NLR ratio despite a low PPV (but a high NPV) and the apparent clinical value of IL-6, TNF, and IL-10 dosing, their relationship with the patient status should be further investigated.

Response:

We appreciate your insightful thoughts. In this initial dataset, we tested the correlation of cytokines alone with various clinical parameters beyond what has already been reported in this version. However, we did not find significant correlations, leading us to report only the significant findings. These results suggest that there may be deeper interactions between these cytokines and the pathophysiological processes, particularly those affecting lung tissue. To further explore these aspects and correlate these cytokines with specific pathological mechanisms, especially in patients who succumbed to the disease, for whom tissue samples are available for analysis, we are currently conducting a continuous follow-up study on COVID-19 patients.

We believe that this continuing research will provide a more comprehensive understanding of the complex relationships between cytokines and clinical outcomes in COVID-19.

Comment 3:

The values of the different markers are somewhat surprising as compared to other studies and the complete data should be presented (as in figure one).

Response:

Thank you for your valuable comments. In response to your suggestion, we have clarified in the manuscript the rationale for presenting only the significant results. Specifically, we have mentioned that of all hematological parameters evaluated, only the NLR at its three stages (admission, endpoint, and highest point) showed significant predictive capabilities for intubation, unlike the other hematological parameters. This information has been added to the text in lines 166-168. Similarly, we clarified that in the analysis of predictive significance for survival, only the NLR showed significant predictive capabilities, which has been added in lines 203-204.

Comment 4:

A multivariate analysis might be of interest.

Response:

Thank you for your insightful comment. We initially performed a Principal Component Analysis (PCA) as an approach to multivariate analysis. However, the results indicated that for the conditions of intubation and survival, the primary components for these groups were the NLR in the three stages analyzed (admission, endpoint, and highest point) as well as all the evaluated cytokines. Therefore, we proceeded with the analyses focused on these significant parameters.

Comment 5:

The authors declare to have performed weekly evaluations of the biological parameters and clinical data of patients and unfortunately they use only partially these data. What is the mean duration stay of the patients in the hospital? Is it possible to have a description of the evolution of these parameters by groups of patients from hospital admission until their discharge (or death)? That would give insight into the relation between the most relevant parameters (among the ones used by the authors) and the outcome of the patients (over several weeks?). It would be useful (although the study is limited to one center) for their follow-up.

Response:

Thank you for your comments. We have added the average duration of hospital stay to the manuscript, which is now found in lines 101 to 102. The mean duration of hospital stay was 21.3 days, reflecting the prolonged care required for severe cases. We agree that a longitudinal analysis of the evolution of parameters over the course of hospitalization would provide valuable insights. This approach is indeed interesting and is part of our ongoing research. Due to the complexity and volume of the data, these analyses are taking longer to complete, and we aim to present them in future publications as we gather more comprehensive data. This ongoing work focuses on correlating cytokine levels and other biomarkers with clinical outcomes over time, which we believe will contribute significantly to understanding the disease's progression and improving patient management. The initial analysis of longitudinal data suggests that a detailed examination of these trends will provide deeper insights, which we are currently pursuing.

Comment 6:

The addresses of the companies which provided the reagents and the laboratory equipment have to be indicated.

Response:

Thank you for your observation. We have added the addresses of the companies that provided the reagents and laboratory equipment. This information can be found in the methods section between lines 118, 120, and 125.

Final Comment:

We appreciate the comments and suggestions from the reviewers, which have greatly contributed to enhancing the quality of our manuscript. Comments 2 through 6 had already been addressed in the initial round of revisions, and we include them again here for completeness in case of any oversight.

We look forward to your feedback and hope that the revisions meet the requirements for publication in your esteemed journal.

Round 3

Reviewer 3 Report

Comments and Suggestions for Authors

I have appreciate the responses to my comment and their taking into account..

I hope that you will  complete this study as promised.

Author Response

Comment:

I have appreciate the responses to my comment and their taking into account..

I hope that you will  complete this study as promised.

Response:

Thank you for your continued engagement and for acknowledging our responses to your previous comments.

We are fully committed to advancing this research as outlined in our responses. We are actively conducting the follow-up studies and analyses discussed previously. The completion of this comprehensive work is anticipated to provide valuable insights, which we plan to submit for consideration in a future publication in “Vaccines.”

Your constructive feedback has been instrumental in guiding our ongoing efforts, and we look forward to the opportunity to contribute further to the field through these upcoming publications in “Vaccines.”

Thank you once again for your encouragement and support.